# A Bridge-Linked Phosphorus-Containing Flame Retardant for Endowing Vinyl Ester Resin with Low Fire Hazard

**DOI:** 10.3390/molecules27248783

**Published:** 2022-12-11

**Authors:** Zihui Xu, Jing Zhan, Zhirong Xu, Liangchen Mao, Xiaowei Mu, Ran Tao

**Affiliations:** 1School of Civil Engineering, Anhui Jianzhu University, Hefei 230601, China; 2State Key Laboratory of Fire Science, University of Science and Technology of China, Hefei 230026, China; 3Anhui Province Key Laboratory of Human Safety, Hefei 230601, China

**Keywords:** vinyl ester resin, flame retardant, thermal stability, mechanism analysis

## Abstract

The high flammability of vinyl ester resin (VE) significantly limits its widespread application in the fields of electronics and aerospace. A new phosphorus-based flame retardant 6,6’-(1-phenylethane-1,2 diyl) bis (dibenzo[c,e][1,2]oxaphosphinine 6-oxide) (PBDOO), was synthesized using 9,10-dihydro-9-oxa-10-phosphaphenanthrene-10-oxide (DOPO) and acetophenone. The synthesized PBDOO was further incorporated with VE to form the VE/PBDOO composites, which displayed an improved flame retardancy with higher thermal stability. The structure of PBDOO was investigated using Fourier transformed infrared spectrometry (FTIR) and nuclear magnetic resonances (NMR). The thermal stability and flame retardancy of VE/PBDOO composites were investigated by thermogravimetric analysis (TGA), vertical burn test (UL-94), limiting oxygen index (LOI), and cone calorimetry. The impacts of PBDOO weight percentage (wt%) on the flame-retardant properties of the formed VE/PBDOO composites were also examined. When applying 15 wt% PBDOO, the formed VE composites can meet the UL-94 V-0 rating with a high LOI value of 31.5%. The peak heat release rate (PHRR) and the total heat release (THR) of VE loaded 15 wt% of PBDOO decreased by 76.71% and 40.63%, respectively, compared with that of untreated VE. In addition, the flame-retardant mechanism of PBDOO was proposed by analyzing pyrolysis behavior and residual carbon of VE/PBDOO composites. This work is expected to provide an efficient method to enhance the fire safety of VE.

## 1. Introduction

Vinyl ester resin (VE) is prepared by the reaction of bisphenol-A epoxy resin and methacrylic acid [1]. Due to its excellent electrical insulation, adhesion, chemical resistance, and easy processing characteristics, VE plays a vital role in industries such as coatings, adhesives, anti-corrosion paints, and electronics [2,3,4,5]. However, the fire hazard potential of VE is higher than that of bisphenol-A epoxy resin due to its easily degradable vinyl [4,5]. The inflammability of VE limits its application in construction, aviation, and other special fields [6,7,8,9]. Therefore, it is imperative to improve the flame-retardant properties of VE.

Phosphorus-based flame retardant has been widely used because of its eco-friendly degradation products and high flame-retardant efficiency [10,11,12,13]. 9,10-dihydro-9-oxa-10-phospha-phenanthrene-10-oxide (DOPO) and its derivatives are often used as an efficient reactive phosphorus-based flame retardant for epoxy resins [14,15,16,17,18]. DOPO can exert flame retardants effect in the gas phase and condensed phase by releasing free radicals and promoting carbonation [19]. The limiting oxygen index (LOI) and vertical burning rating of epoxy resin can reach 36.2% and V-0 after incorporating 3 wt% ABD, which is a flame retardant synthesized by acrolein and DOPO [15]. Fang et al. synthesized a new flame retardant TDA containing phosphorus, nitrogen, and silicon-based on DOPO, and 25 wt% TDA can endow EP composites with an LOI value of 33.4% [18]. However, the flame retardancy of VE with DOPO and its derivatives is low, and the LOI of VE composites can only reach 31.5% with 30 wt% DOPO-2-hydroxylethyl acrylates [16].

The main factors that affect the flame-retardant efficiency of DOPO are: (a) Phosphorus content of DOPO derivatives. (b) Other functional groups in DOPO derivatives [17,19]. Epoxy resin with 17.5 wt% phenethyl-bridged DOPO derivative (PN-DOPO) achieved vertical burn test (UL-94) V-0 grade, and its average heat release rate (av-HRR) value was reduced by 32.5% [20]. Acetophenone can react with DOPO to construct a bridging structure between DOPO molecules, thus increasing the overall content of phosphorus and aromatic group in the formed DOPO derivatives [19,20]. Therefore, acetophenone was considered as a sensitizer for DOPO to generate a novel DOPO derivative with improved flame-retardant efficiency.

In this work, a novel phosphorus-based flame retardant named PBDOO was synthesized by DOPO and acetophenone. The thermal stability, mechanical properties, and flame-retardant properties of VE composites with PBDOO were studied in depth. Finally, the flame-retardant mechanism of VE composites was further revealed.

## 2. Result and Discussion

### 2.1. Characterization of PBDOO

The Fourier transform infrared (FTIR) spectra of PBDOO is shown in Figure 1a, and the groups corresponding to each absorption peak are listed in Table 1.

It is well-known that the P-H vibration absorption peak at 2400 cm^−1^ is the most obvious characteristic absorption peak of DOPO [21,22]. And acetophenone has a C=O absorption peak of carbonyl around 1727 cm^−1^. Neither of these absorption peaks is shown in Figure 1a, indicating that the target product is free of DOPO and acetophenone [21]. The new absorption peaks listed in Table 1 indicate that the target product is successfully synthesized [22,23].

^1^H and ^31^P nuclear magnetic resonance (NMR) spectra were used to verify the structure of PBDOO. As shown in Figure 1b,c, the characteristic peak at 2.85 ppm in the ^1^H NMR spectrum of PBDOO corresponds to the hydrogen atom in the -CH_2_- structure [21,22,24]. In addition, the typical peak at 6.8–8.0 ppm corresponds to the hydrogen atom of the benzene ring [21]. ^31^P NMR is multimodal at 34.0–37.0 ppm. The presence of these peaks indicates that PBDOO is successfully synthesized.

Thermogravimetric analysis (TGA) and Thermogravimetric differentiation analysis (DTG) curves of PBDOO under nitrogen and air atmospheres are shown in Figure 2. The corresponding data are displayed in Table 2. Among them, T_5%_ is defined as the temperature at 5% weight loss, T_max_ is defined as the temperature at which the maximum weight loss rate is, and the temperature selected for the char residues is 800 °C.

As shown in Figure 2a, under air atmosphere, the T_5%_ of PBDOO is 344 °C, the T_max_ of PBDOO is 388 °C, and its char residues at 800 °C are 7.2%. Figure 2b shows the results under a nitrogen atmosphere. The T_5%_ and T_max_ are 352 °C and 405 °C, respectively. It can be seen that PBDOO has high thermal stability in both air and nitrogen atmosphere, and the initial decomposition temperatures are above 340 °C. The processing temperature of VE is about 150 °C, so PBDOO can meet the processing conditions of VE. Furthermore, the amount of char residue under the air atmosphere is significantly higher than that under a nitrogen atmosphere, which may be because the presence of oxygen promotes the formation of stable carbon layers [24,25].

### 2.2. Morphologies and Mechanical Properties of the VE Composites

In order to study the effect of PBDOO on the mechanical properties of VE materials, the tensile strength and elongation at break of VE composites with different proportions of flame retardant were tested. The test results are shown in Figure 3. The tensile strength and elongation at the break of the VE without PBDOO are 33 MPa and 25%, respectively. As the content of PBDOO increases, the strength of the cured product first increases from 33 MPa to 36 MPa, and then dropped to 14 MPa for VE-15. The elongation at break changed slightly from 25% of VE-0 to 21% of VE-15. The result indicates that with a small amount of PBDOO involved in the curing process, the VE composites can form a stable cross-linked structure, which gives an increase in tensile strength [26]. When the content of PBDOO further increases, the steric hindrance produced by the rigid DOPO group limited the cross-linking between VE and the curing agent. So the tensile strength and elongation at the break decrease [26,27].

Figure 4 shows the scanning electron microscope (SEM) image of the cross-section of VE-0 and VE-15. It could be seen that the surface of VE-15 is slightly rougher than that of VE-0, which indicates a minor effect of PBDOO on VE. In addition, there are no noticeable large particles on the fracture surfaces of the two samples, which also indicates that PBDOO could be well dispersed in the VE material. This result shows that PBDOO has good compatibility with VE and can be well dispersed in VE composites.

### 2.3. Thermal Stability

TGA and DTG curves of the VE composites under nitrogen and air atmospheres are shown in Figure 5 and the corresponding data are summarized in Table 2. As shown in Figure 5a,c, with the addition of PBDOO, the T_5%_ temperature of the VE composites moves forward slightly, indicating that the addition of the flame retardant reduces the initial decomposition temperature of VE.

It could be observed from Figure 5b that all the samples have a two-stage decomposition process in the air atmosphere. The first decomposition occurs at around 330 °C and reaches T_max_ at 410 °C, which may be due to the decomposition of aromatic rings and alkyl chains [23]. The second degradation of VE composites occurs at about 500 °C and reaches T_max_ at about 550 °C, which may be due to the further thermal-oxidative decomposition of the unstable char layer at high temperatures. The results in Table 2 show that as the amount of PBDOO increases, so does the residual carbon of the VE composites.

### 2.4. Flame-Retardant Properties of the VE Composites

LOI and UL-94 were used to judge the flame retardancy of the samples, and the data are shown in Table 3. As the content of PBDOO increases, the LOI value of VE composites also increases significantly. The LOI value of VE-15 reached 31.5%, which is much higher than that of VE-0. And the vertical burning rating of VE got V-0 after incorporating 15 wt% PBDOO. Therefore, it could be concluded that PBDOO can effectively improve the flame-retardant properties of VE composites.

The combustion behavior of VE was further studied by a cone calorimeter. The total heat release (THR) and heat release rate (HRR) curves of VE composites are shown in Figure 6. Some of the essential parameters are summarized in Table 4. From Figure 6a,b, it can be seen that VE-0 burns quickly and releases a large amount of heat after being ignited. The peak of HRR (PHRR) and THR of VE-15 are 339 kW/m^2^ and 47.9 MJ/m^2^, respectively, which are reduced by 76.71% and 40.36% compared with those of VE-0.

Figure 7 shows the changes in CO production rate (COPR) and CO_2_ production rate (CO_2_ PR) of the VE material during the cone tests. The peak value of CO_2_ PR for VE-0 is 1.12 g/s. With the increase of PBDOO content in VE composites, the peak values of CO_2_ PR decreased by 44.64%, 74.11%, and 80.35%, respectively. But the values of COPR have no obvious change trend as observed in Figure 7b. The changes in CO_2_ PR may be result by the thermal decomposition of PBDOO. The decomposition products can quench the active pyrolysis fragments from VE and inhibit the combustion, which reduces the generation of CO_2_ [25,28,29].

As seen in Table 3, the addition of PBDOO increases the time to ignition (TTI) of VE composites. The TTI of VE-0 and VE-15 were 59 s and 67 s, respectively, which indicated that the incorporation of highly stable flame retardants plays a good role in physical protection.

### 2.5. Flame-Retardant Mechanism

#### 2.5.1. Condensed Phase Analysis

Figure 8a–c show the photographs of the VE composites before and after heating at 650 °C for 5 min in a muffle furnace. It could be seen from Figure 8a that there is almost no change in the appearance of VE composites after the addition of PBDOO, which is yellow-brown and transparent. The result in Figure 8b shows that VE-0 has no residue after calcination in the muffle furnace, while VE-15 has a noticeable carbon residue, which is consistent with the thermogravimetric test results. This is probably because the presence of aromatic rings and phosphorus in PBDOO promotes the formation of carbon layers, and therefore more residual carbon is obtained [24,25]. It could be seen from the magnification in Figure 8c that the carbon layer with an expanded three-dimensional structure is loose, porous, and fragile. In order to better characterize the microstructure of the carbonized layer, the carbon residue is observed by SEM, as shown in Figure 8d. It could be seen that the carbon residue is porous, and most of the tiny pores on the surface are cracked. This may be due to a large amount of gas rushing out of the carbon layer during decomposition, which is consistent with the phenomenon observed in the UL-94 test.

#### 2.5.2. Pyrolysis Behaviors of the VE Composites

To better study the degradation process and flame-retardant mechanism of PBDOO in VE, the thermogravimetric analysis-infrared spectroscopy (TG-IR) of VE-0 and VE-15 under nitrogen gas conditions was investigated. Figure 9a shows the FTIR spectra of the pyrolysis gaseous products of VE and its composites at the maximum decomposition rate. The FTIR curve of the VE composites with 15 wt% PBDOO is similar to that of VE-0, and some representative pyrolysis gaseous products are observed. The peaks near 1400–1600 cm^−1^ and 650–900 cm^−1^ correspond to aromatic compounds. The peaks at 1000–1300 cm^−1^ are mainly absorption peaks of C-O compounds. The absorption peaks centered at 1730 cm^−1^ and 2306 cm^−1^ are mainly absorption peaks of carbonyl groups containing aldehydes and CO_2_ [23,25,26,27]_._ It is worth noting that the peak strength of the VE composites is significantly reduced after the addition of PBDOO.

The intensity curves of total pyrolysis products are plotted in Figure 9b. It can be seen that the absorption strength of the VE composites with flame retardant is significantly lower than that of the neat samples, which means that fewer gaseous products are detected for VE-15.

To better understand the differences between the gaseous products of VE and VE-15, the absorption intensities of some typical pyrolysis products are shown in Figure 10. The intensities of three absorption peaks of VE-15 at 829 cm^−1^, 1509 cm^−1^, and 1610 cm^−1^ are lower than those of VE-0. This result demonstrates that the introduction of PBDOO significantly reduces the production of aromatic compounds, which can participate in the combustion process and release more heat. Therefore, adding PBDOO can effectively increase the fire safety performance of VE composites.

The appearance of the C=O absorption peak at 1732 cm^−1^ in Figure 10b may be produced by the decomposition of the ethyl chain segment. The decrease in its intensity also indicates that the decomposition of VE composites is suppressed and the thermal stability is improved after the addition of flame retardant. The hydrocarbon absorption peak curve in Figure 10d also indicates the inhibitory effect of PBDOO on the decomposition of VE composites. Figure 10c shows the absorption intensities of CO_2_ at around 2306 cm^−1^, and it can be seen that the curve of VE-15 is much lower than that of VE-0. This may be due to the decrease in the number of HO· radicals, which is caused by the quenching effect of free radicals are generated from the degradation of PBDOO [30].

As a phosphorus-containing flame retardant, PBDOO mainly functions in gaseous and condensed phases. According to TG data, PBDOO can reduce the initial decomposition temperature of VE composites, which is conducive to the earlier formation of a stable carbon layer to isolate external air and heat. And the decomposition of chemical bonds can release DOPO groups, which are further cleaved to form free radicals such as PO· and PO_2_·. These phosphorus-containing radicals can capture active radicals in the flame area, inhibit the chain reaction, and interrupt or slow the combustion of VE composites [30,31]. The DOPO groups in the matrix that have not been cleaved to form radicals and the quenched productions of phosphorus-containing radicals can further aggregate at high temperatures to form phosphorus-containing residues [27]. This residue layer can somewhat prevent the release of combustible gases and heat [32]. The cone calorimeter results show that the HRR and THR values of VE-0 are significantly higher than these of VE-15, which also verifies that PBDOO can suppress the heat release of VE composites during the combustion process. Therefore, in conclusion, the inhibition effect of PBDOO on combustible gas and the formation of condensed phase carbon layer together play a good flame-retardant effect on VE composites. It can be simplified in Figure 11.

## 3. Experimental

### 3.1. Materials

9, 10-dihydro-9-oxa-10-phosphaphenanthrene-10-oxide (DOPO), acetophenone, xylene, phosphorus oxychloride, isopropanol, acetone and dibenzoyl peroxide (BPO) were purchased from Aladdin Co., Ltd. (Shanghai, China). Deionized water was self-produced in the laboratory. All chemicals were used as received without any special treatment.

### 3.2. Synthesis of PBDOO

As shown in Figure 12, a novel phosphorus-containing flame retardant named PBDOO was synthesized by the addition reaction between DOPO and acetophenone. A certain amount of DOPO, acetophenone, and xylene was added to the three-necked flask, and phosphorus oxychloride was slowly added dropwise after being heated to 154 °C under a nitrogen atmosphere. Fractions were collected in a trap at 155 °C. After the dropwise addition of phosphorus oxychloride, stirring was continued for half an hour to make the reaction sufficient. Isopropanol was added to the cooled mixture. The precipitated product was filtered with suction and washed with isopropanol and deionized water to obtain a solid white powder, which was then thoroughly dried at 110 °C.

### 3.3. Preparation of the VE Composites

A series of VE composites with different PBDOO content were prepared. The specific formulations are listed in Table 3. BPO is the curing agent.

A certain amount of VE was added into a mixed system of PBDOO and acetone. The mixture was stirred continuously at 60 °C until acetone was volatilized entirely. PBDOO would dissolve into VE during the stirring process, forming a dark brown transparent viscous mixture. Subsequently, the mixture was stirred rapidly for 15 min at 100 °C. After that, a certain amount of initiator was added to the mix and stirred vigorously for 15 s. The resulting mixture was quickly poured into a mold prepared in advance, transferred to the oven, and cured for 2 h at 100 °C, 130 °C, and 150 °C. After cooling to room temperature, the VE composites were taken out of the mold to obtain samples with different PBDOO content.

### 3.4. Characterization

Instruments Nicolet 6700 Fourier Transform Infrared Spectrometer (Waltham, Massachusetts, USA) was used to record the FTIR spectrum of the sample, and the test wavenumber range was 500–4000 cm^−1^. The ^1^H and ^31^P NMR spectra of the samples were performed on an AVANCE-400 NMR (Bruker company in Zurich, Switzerland) spectrometer, using CDCl_3_ as the solvent and running in Fourier transform mode. The tensile test was done on the YF-900 computerized tensile testing machine (Jiangsu, China) according to ASTM D3039-08. The sample size was 100 mm × 10 mm × 3 mm, and at least 5 parallel samples were tested for each ratio. Scanning electron microscope (SEM), using JEOL JSM-6700F scanning electron microscope (Tokyo, Japan), the voltage used in the microscope is 10 kV. Before the SEM measurement, the sample was sputtered with a thin layer of gold. Thermogravimetric analysis (TGA) test was carried out on the Q5000 thermal analyzer (TA company in Newcastle, Delaware, USA), measured from 25 °C to 800 °C at a heating rate of 20 °C/min under N_2_ and Air atmosphere. The cone calorimeter test was tested on the FTT cone calorimeter (UK) using the ISO 5660-1 standard, the sample (size 100 mm × 100 mm × 3 mm) was placed in the aluminum foil, and the heat radiation flux was 35 kW/m^2^. Three samples for each ratio were tested. LOI was based on the GB/T 2406.2-2009 standard to evaluate the limiting oxygen index (LOI) value on the HC-2 oxygen index analyzer (Jianning, China). The sample size was 100 mm × 10 mm × 3 mm, and 15 samples were taken from each group. According to the GB/T 2408-2008 standard, the vertical combustion test of the sample was carried out on the CFZ-2 instrument (Jiangning, China). The sample size was 130 mm × 13 mm × 3 mm. Thermogravimetric-infrared spectroscopy (TG-FTIR), measured by PerkinElmer (Waltham, Massachusetts, USA) STA8000 thermogravimetric analyzer connected to PerkinElmer Frontier FT-IR spectrometer, the heating rate was 20 °C/min, the atmosphere was N_2_, the flow rate was 45 mL/min. The chamber and transfer tube were kept at 300 °C and 280 °C, respectively.

## 4. Conclusions

In this work, a phosphorous flame retardant based on DOPO was successfully synthesized, characterized, and applied to VE composites. The research indicated that the introduction of PBDOO effectively improved the flame retardancy of VE and reduced its heat release rate of VE during its combustion process. With the addition of 15 wt% PBDOO, the VE composite passed the V-0 level of the UL-94 test and the LOI value reached 31.5%. Compared with neat VE, the PHRR and THR of VE-15 decreased by 76.71% and 40.36%, respectively. The results of TG-IR demonstrated that the added PBDOO significantly reduced the production of aromatic compounds and inhibited the release of combustible gases. In addition, the adoption of PBDOO decreased the tensile strength of VE composites, while the elongation at the break did not change much. Mechanistic analysis shows that adding PBDOO is conducive to forming stabilized carbon layer, which plays an essential role in the early isolation of air and heat. Furthermore, the phosphorus-containing radicals generated by the cleavage of DOPO groups can quickly capture the free radicals in the flame region, delaying or interrupting the combustion. The flame retardant designed by this work is expected to further broaden the application of VE composites in construction, aerospace, and other fields.

## Figures and Tables

**Figure 1 molecules-27-08783-f001:**
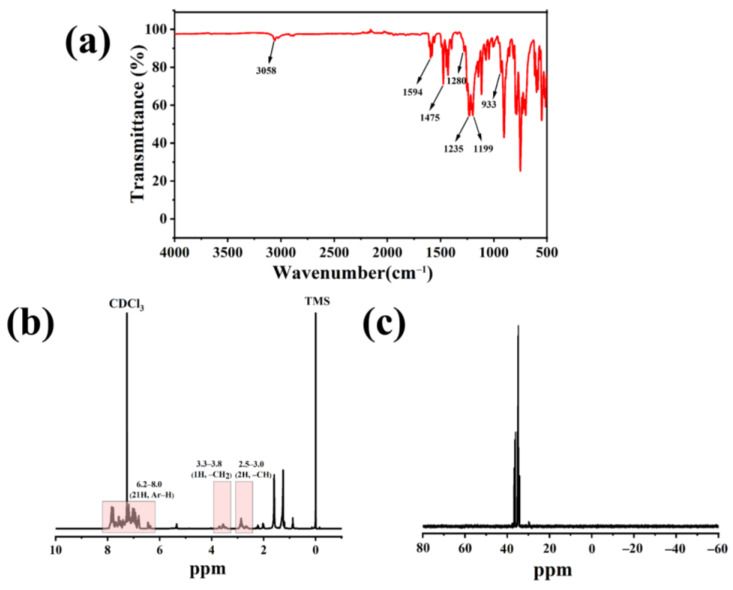
(**a**) FTIR spectra, (**b**) ^1^H NMR spectra and (**c**) ^31^P NMR spectra of PBDOO.

**Figure 2 molecules-27-08783-f002:**
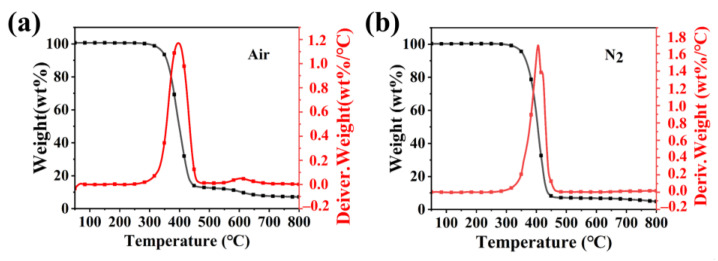
TGA and DTG curves of PBDOO in (**a**) Air and (**b**) N_2_.

**Figure 3 molecules-27-08783-f003:**
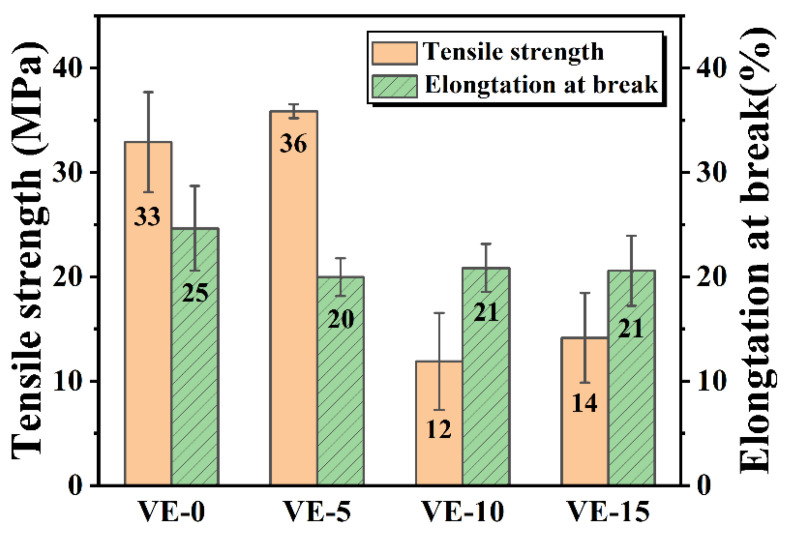
Strength and elongation at break of VE composites.

**Figure 4 molecules-27-08783-f004:**
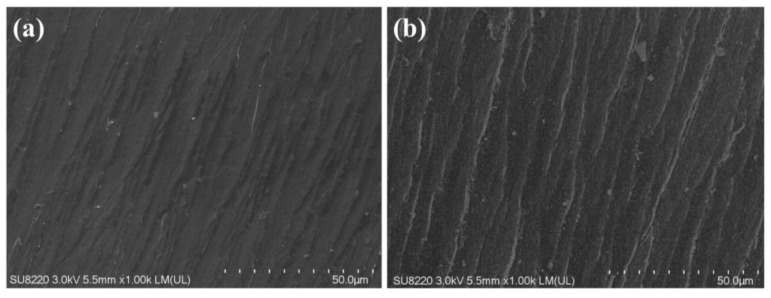
SEM images of the fractured surface of VE composites: (**a**) VE-0, (**b**) VE-15.

**Figure 5 molecules-27-08783-f005:**
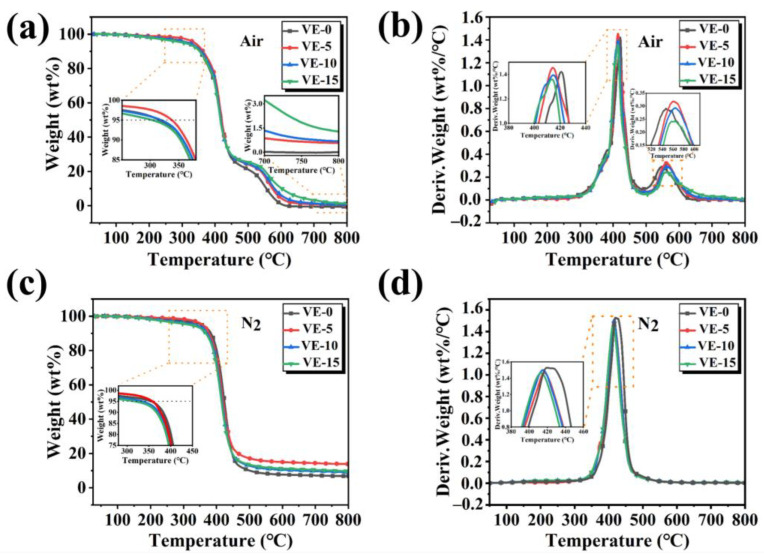
TGA and DTG curves of VE composites in air (**a**,**b**) and N_2_ (**c**,**d**).

**Figure 6 molecules-27-08783-f006:**
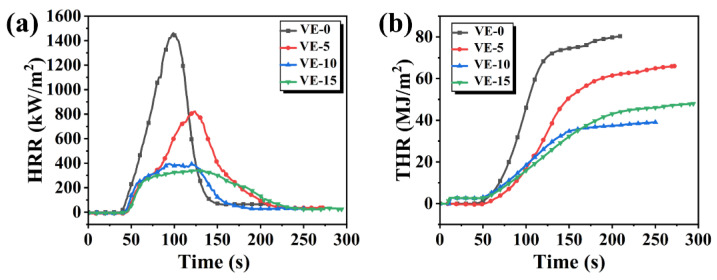
(**a**) Heat release rate of VE composites, (**b**) total heat release of VE composites.

**Figure 7 molecules-27-08783-f007:**
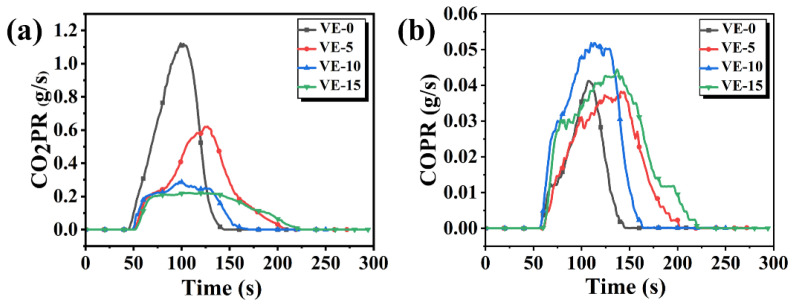
(**a**) CO_2_ production rate of VE composites, (**b**) CO production rate of VE composites.

**Figure 8 molecules-27-08783-f008:**
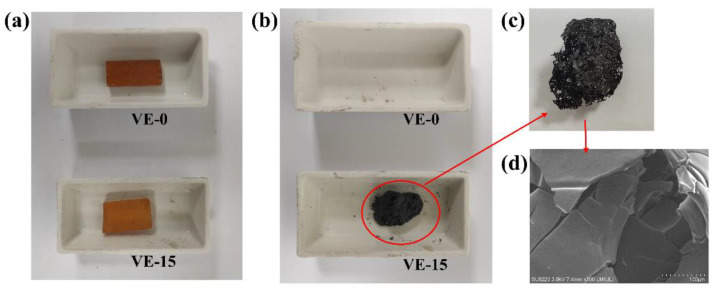
Digital photographs of VE-0 and VE-15 composites (**a**) before and (**b**) after being heated at 650 °C in air; (**c**) Photograph of char carbon of VE-15; (**d**) SEM of the char carbon of VE-15.

**Figure 9 molecules-27-08783-f009:**
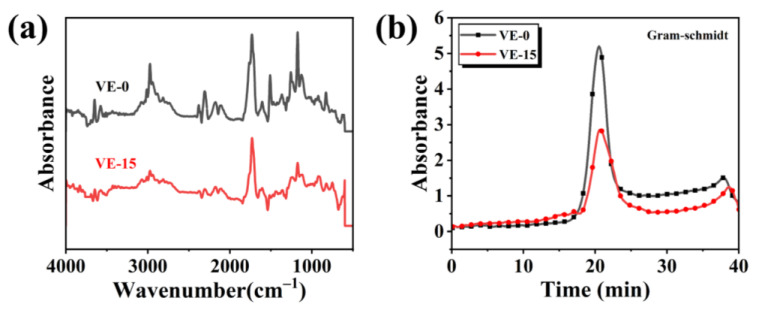
(**a**) FTIR spectra of pyrolysis gaseous products originated from VE-0 and VE-15 at the maximum evolution rate; (**b**) Absorbance intensities of VE-0 and VE-15.

**Figure 10 molecules-27-08783-f010:**
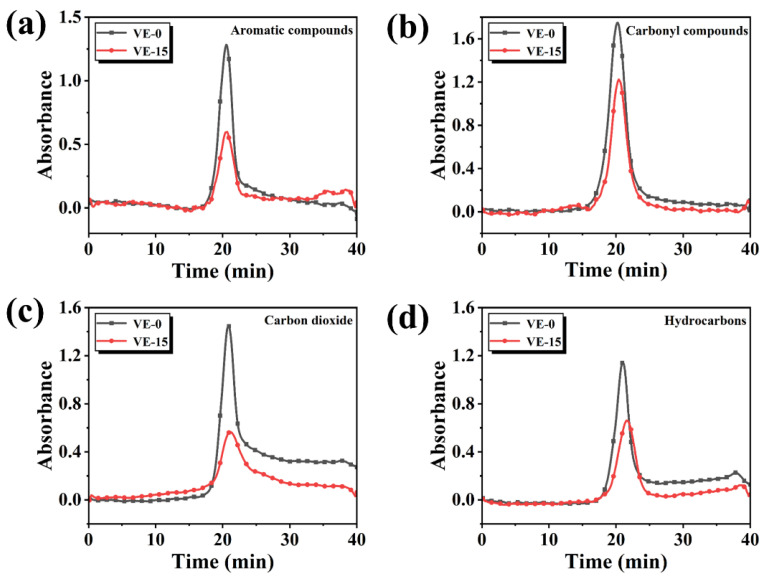
Absorbance intensities of typical pyrolysis products for VE-0 and VE-15: (**a**) Aromatic compounds, (**b**) Carbonyl compounds, (**c**) Carbon dioxide, (**d**) Hydrocarbons.

**Figure 11 molecules-27-08783-f011:**
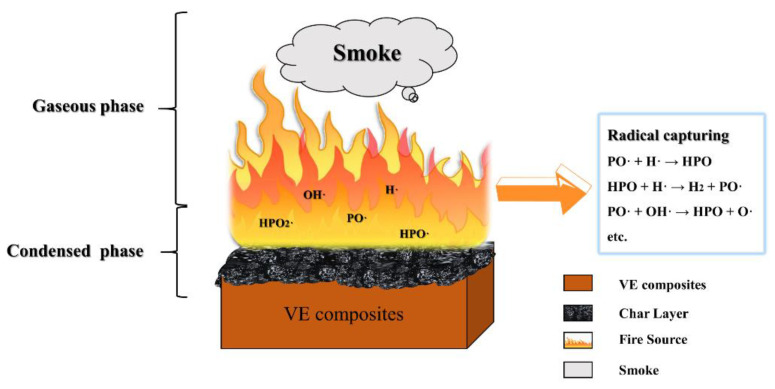
Flame retardant mechanism of PBDOO.

**Figure 12 molecules-27-08783-f012:**
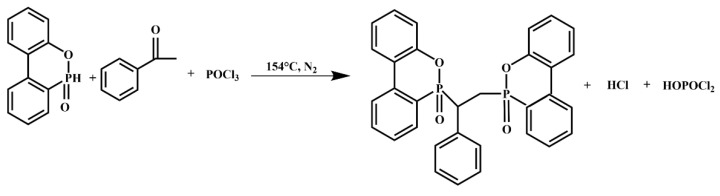
Synthesis route of PBDOO.

**Table 1 molecules-27-08783-t001:** Wavenumber and corresponding chemical groups in PBDOO.

Wavenumber (cm^−1^)	Corresponding Chemical Groups
3058	C-H stretching vibration peak of the aromatic ring
1594	P-Ar stretching vibration peak
1475	P-Ph absorption peak
1280	O=P-O absorption peak of the benzene ring structure
1235	P=O vibration absorption peak
1199, 933	P-O-Ph vibration absorption peak

**Table 2 molecules-27-08783-t002:** TGA and DTG data of PBDOO and the VE composites.

Sample	N_2_			Air			
	T_5%_	T_max_	Residue	T_5%_	T_max1_	T_max2_	Residue
	(°C)	(°C)	(wt%)	(°C)	(°C)	(°C)	(wt%)
PBDOO	352	405	4.9	344	388	--	7.2
VE-0	316	420	6.8	356	422	546	0.0
VE-5	338	414	13.8	358	414	561	0.6
VE-10	319	414	9.1	334	415	562	0.7
VE-15	304	414	9.7	317	413	561	1.3

**Table 3 molecules-27-08783-t003:** LOI and UL-94 results of the VE composites.

Sample	VE (g)	BPO (g)	PBDOO (g)	LOI (%)	UL-94
VE-0	50.00	1.50	0.00	22.00	V-2
VE-5	47.50	1.43	2.50	24.50	V-1
VE-10	45.00	1.35	5.00	25.50	V-1
VE-15	42.50	1.28	7.50	31.50	V-0

**Table 4 molecules-27-08783-t004:** Cone calorimetry data of VE composites.

Sample	TTI	PHRR	THR	CO_2_ PR	COPR
	(s)	(Kw/m^2^)	(MJ/m^2^)	(g/s)	(g/s)
VE-0	59	1455	80.4	1.12	0.041
VE-5	66	822	66.0	0.62	0.038
VE-10	64	385	39.1	0.29	0.052
VE-15	67	339	47.9	0.22	0.044

Note: CO_2_ PR value in Table 4 is the peak of the CO_2_ production rate; COPR value in Table 4 is the peak of the CO production rate.

## Data Availability

Not applicable.

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
