# Peer review of "A Bridge-Linked Phosphorus-Containing Flame Retardant for Endowing Vinyl Ester Resin with Low Fire Hazard"

_molecules, 2022, doi:10.3390/molecules27248783_

Round 1

Reviewer 1 Report

In this paper, a new phosphorus-based flame retardant PBDOO was synthesized and applied to VE composites to improve its flame retardant properties and reveal its mechanism. But there are still some questions in this paper. Therefore, we recommend that this manuscript be published after minor revision.

1. Figure 2 is a bit blurry in the paper. It is recommended to enlarge or separate the FTIR and NMR spectra of PBDOO.

2. It would be better if one or two examples of DOPO derivatives reducing the fire hazard of VE and EP were added in the second paragraph of the introduction.

3. Add reaction conditions at the arrows in Figure 1 to make the figure more visual and transparent.

4. The conclusion does not express the significance and value of the practical application. It is suggested to supplement one sentence to give a brief description.

5. The sentence in lines 270-274 is hard to understand and should be considered for revision.

Reviewer 2 Report

Dear editor,

The manuscript entitled "A bridge-linked phosphorus-containing flame retardant for endowing vinyl ester resin with low fire hazard" reported a new phousphrous compound and investigated systematically its flame retardant properties as additives to the vinyl ether resins. The research has novelity by the new structure, so I recommend for acceptance of publication after addressing the following points:

1. The 1H-NMR spectrum should be clear enough to identify peak assignemnt for PBDOO.

2. If possible, the MS spectrum for PBDOO should be also provided for structure confirmation.

3. The reaction route may be modified to show where the choride goes after reaction (Figure 1).
